# Investigating Modifiable Risk Factors Across Dementia Subtypes: Insights from the UK Biobank

**DOI:** 10.3390/biomedicines12091967

**Published:** 2024-08-31

**Authors:** Xiangge Ma, Hongjian Gao, Yutong Wu, Xinyu Zhu, Shuicai Wu, Lan Lin

**Affiliations:** Department of Biomedical Engineering, College of Chemistry and Life Science, Beijing University of Technology, Beijing 100124, China; maxiangge@emails.bjut.edu.cn (X.M.); gaohongjian@bjut.edu.cn (H.G.); wyt191026@emails.bjut.edu.cn (Y.W.); zhuxyu@emails.bjut.edu.cn (X.Z.); wushuicai@bjut.edu.cn (S.W.)

**Keywords:** dementia, Cox analysis, risk factors, risk domains, population attributable fractions

## Abstract

This study investigates the relationship between modifiable risk factors and dementia subtypes using data from 460,799 participants in the UK Biobank. Utilizing univariate Cox proportional hazards regression models, we examined the associations between 83 modifiable risk factors and the risks of all-cause dementia (ACD), Alzheimer’s disease (AD), and vascular dementia (VD). Composite scores for different domains were generated by aggregating risk factors associated with ACD, AD, and VD, respectively, and their joint associations were assessed in multivariable Cox models. Additionally, population attributable fractions (PAF) were utilized to estimate the potential impact of eliminating adverse characteristics of the risk domains. Our findings revealed that an unfavorable medical history significantly increased the risk of ACD, AD, and VD (hazard ratios (HR) = 1.88, 95% confidence interval (95% CI): 1.74–2.03, *p* < 0.001; HR = 1.80, 95% CI: 1.54–2.10, *p* < 0.001; HR = 2.39, 95% CI: 2.10–2.71, *p* < 0.001, respectively). Blood markers (PAF = 12.1%; 17.4%) emerged as the most important risk domain for preventing ACD and VD, while psychiatric factors (PAF = 18.3%) were the most important for preventing AD. This study underscores the potential for preventing dementia and its subtypes through targeted interventions for modifiable risk factors. The distinct insights provided by HR and PAF emphasize the importance of considering both the strength of the associations and the population-level impact of dementia prevention strategies. Our research provides valuable guidance for developing effective public health interventions aimed at reducing the burden of dementia, representing a significant advancement in the field.

## 1. Introduction

Dementia is a complex neurodegenerative syndrome characterized by a gradual decline in cognitive functions that severely disrupts daily activities [1,2,3]. With the global population aging and mortality rates declining among younger age groups, the prevalence of dementia is anticipated to escalate notably in the coming decades. This syndrome affects a wide range of cognitive abilities, including memory, thinking, behavior, and functional abilities [4]. Alzheimer’s disease (AD) is the most common form, responsible for 60% to 80% of dementia cases [5,6,7]. Other major types include vascular dementia (VD), resulting from cerebrovascular damage [8]; frontotemporal dementia, associated with the degeneration of the frontal and temporal lobes [9]; and Lewy body dementia, characterized by abnormal protein deposits in the Lewy bodies [10]. As the global population ages, the prevalence of dementia is anticipated to escalate significantly, posing substantial challenges to healthcare systems, economies, and families [11,12]. The World Health Organization projects that the number of individuals living with dementia could triple by 2050, reaching around 152 million. This trend highlights the urgent need for further research into dementia’s etiology, risk factors, and progression [13,14]. Particularly, modifiable conditions such as hypertension [15], diabetes [16], obesity [17], and smoking [18] have been recognized as significant risk factors, highlighting the importance of preventive healthcare measures.

Epidemiological research is crucial for understanding dementia’s distribution and determinants. Evidence suggests that individuals with low to moderate genetic risk profiles who maintain favorable modifiable risk factors have a lower risk of dementia [19]. Environmental factors like air pollution, particularly traffic-related pollution, are emerging as significant risk factors for both VD and AD [20]. Furthermore, atrial fibrillation has been linked to accelerated cognitive decline and an evaluated risk of dementia [21]. Psychiatric factors, such as a history of depression, also contribute to dementia risk, though depression may often act as an early marker rather than a standalone risk factor [22]. Despite these insights, understanding the factors associated with dementia and their relative importance can vary significantly across different cohorts. When sample sizes are small, there is a risk of limited statistical power, potentially restricting the ability to generalize findings to larger populations. Larger, robust studies are essential to confirm these associations and clarify the complex interplay of genetic, environmental, and lifestyle factors in dementia development.

The UK Biobank (UKB) [23,24,25] is a groundbreaking biomedical database housing extensive genetic, lifestyle, and health data from over 500,000 volunteers across the United Kingdom. Studies utilizing the UKB have significantly advanced our understanding of dementia risk. In older adults without cognitive impairment, adverse lifestyle factors and high genetic risk are linked to increased dementia risk. Key factors include irregular physical activity, poor diet, excessive alcohol consumption, smoking [26,27], and longer sleep duration (>9 h) [28]. Social isolation, but not loneliness, has been found to increase the risk of dementia across all genetic risk levels [29]. Blood biomarkers, such as vitamin D deficiency and altered hormone levels, as well as reduced kidney function, are associated with dementia risk [30]. Grip strength reduction by 5 kg is also linked to dementia [31]. However, most studies have focused on single or specific factors without examining the interactions between them. Stevie Hendriks et al. included 39 potential risk factors in a Cox model and found that 15 factors were significantly associated with higher young-onset dementia risk, including sociodemographic factors and psychiatric factors [32]. Zhang et al. assessed the combined relationship of 210 modifiable risk factors with dementia using a multivariable Cox model and estimated that removing adverse profiles in risk domains could prevent up to 47.0–72.6% of dementia cases, using population attributable fractions (PAF) [33]. However, dementia represents a heterogeneous spectrum, encompassing various subtypes, each characterized by distinct pathological mechanisms and biological features. AD, for instance, is defined by beta-amyloid plaques and neurofibrillary tangles, while VD is associated with cerebrovascular damage. This diversity highlights the limitation of studying dementia risk in a generalized manner, as it may not fully capture the specific factors contributing to each subtype. While certain risk factors may influence multiple dementia subtypes, individual subtypes can possess unique modifiable risk factors that warrant targeted investigation. For example, hypertension and diabetes are more directly associated with VD. This underscores the need for subtype-specific studies to better understand unique risk factors and develop targeted prevention and intervention strategies.

To fill the existing knowledge gap, we conducted a prospective study using the UKB data, a comprehensive resource that includes genetic, lifestyle, and health data from over half a million participants. The primary objective of this study was to explore the associations between multiple risk factors and all-cause dementia (ACD), as well as its specific subtypes, AD and VD. To achieve this, we first thoroughly identified the impact of various individual risk factors on dementia and its subtypes. Subsequently, we created composite scores across multiple risk domains. By employing these composite scores, we were able to investigate the collective influence of a range of risk factors on ACD and its specific subtypes, thereby enhancing our understanding of the complex interactions among multidimensional risk factors. Another critical aspect of our study was the calculation of the PAF for each risk domain, as well as the overall PAF. The calculation of PAF provided us with a quantitative tool to estimate the proportion of dementia cases, particularly those of the AD and VD subtypes, that could be linked to modifiable risk factors. By extrapolating the PAF, we aimed to uncover the potential preventive effects of interventions targeting these risk factors, thereby offering insights into the impact of such preventive measures on the incidence of dementia and its subtypes.

## 2. Materials and Methods

### 2.1. Study Population

This study utilized data from the UKB. Launched in 2006, the UKB aims to improve the prevention, diagnosis, and treatment of a wide range of serious and life-threatening illnesses, including dementia. The vast dataset includes detailed information on participants’ physical measures, biological samples, and health records, making it one of the most extensive and valuable resources for medical research. Written informed consent was obtained electronically from all participants, and the study received ethical approval from the North West Multi-Center Research Ethics Committee. For this analysis, individuals with dementia at baseline, those with more than 20% missing data, and those with missing covariates were excluded. The research was conducted under UKB application number 68382.

### 2.2. Dementia Diagnosis

This study investigated the incidence of ACD, including its subtypes AD and VD. To identify dementia events, data from the UKB fields, primarily spanning from 42,018 to 42,025, was utilized. Information was sourced from the English Hospital Episode Statistics, Scottish Morbidity Records, and the Welsh Patient Episode Database. The classification of dementia types adhered to the International Classification of Diseases, Ninth Revision (ICD-9), and Tenth Revision (ICD-10). Participants were followed until the occurrence of the earliest of the following events: initial diagnosis, date of death, loss of follow-up, or the last date of available information (December 2020).

### 2.3. Modifiable Factors

Baseline data were extracted from the UKB, and variables with more than 25% missing values were excluded from the analysis. After data processing, 51 potential risk factors were identified and categorized into seven domains: (1) sociodemographic factors: including socioeconomic status, education, and employment status; (2) lifestyle factors: including drinking status, smoking status, physical activity, diet, sleep, social isolation, and loneliness; (3) environmental factors: including nitrogen oxides air pollution, particulate matter air pollution, greenspace percentage, water percentage, and natural environment percentage; (4) physical measures: including handgrip strength, systolic blood pressure, diastolic blood pressure, body fat percentage, trunk fat percentage, etc.); (5) blood marker factors: including estimated glomerular filtration rate (eGFR), vitamin D, C-reactive protein, albumin, insulin-like growth factor 1 (IGF-1), etc.; (6) medical history: including stroke, hypertension, diabetes, atrial fibrillation, coronary artery disease, etc.; and (7) psychiatric factors: including depression, anxiety, bipolar disorder, sleep problems, alcohol use disorder, etc. Physical activity scores were calculated based on the American Heart Association recommendations [26]. Diet scores were computed using dietary priorities for cardiometabolic health [27]. The sleep pattern was assessed using five questions [34]. The social isolation scale consisted of three questions, and the loneliness scale comprised two questions [29]. eGFR was calculated using the CKD-EPI (Chronic Kidney Disease Epidemiology Collaboration) creatinine equation [35]. Missing values in the variables were imputed using random forest algorithms. Categorical variables were simplified into binary or ternary categories, while continuous variables were divided into three categories based on tertiles. After subdivision, we ultimately identified 83 modifiable risk factors. The detailed calculation and classification process for these variables is provided in Appendix A. 

### 2.4. Statistical Analyses

To explore the relationships between these risk factors and the incidence of dementia, we employed Cox proportional hazards regression models. This approach enabled us to calculate the “hazard ratio” (HR) for each risk factor, thereby quantifying the impact of a specific risk factor on the likelihood of developing dementia. To address the issue of multiple comparisons, a Bonferroni correction was applied, adjusting the significance threshold to 0.0006024 (0.05 divided by 83 tests). This adjustment was crucial for controlling the false positive rate and ensuring the reliability of our findings. The associations were adjusted for baseline age and sex to account for potential confounding factors. The proportional hazards assumption, a key requirement of the Cox model, was verified using Schoenfeld residuals, with variables showing *p* > 0.001 considered to satisfy this assumption. To further ensure the robustness of our model, multicollinearity among the variables was assessed using variance inflation factors (VIFs). Multicollinearity can impede the model’s ability to accurately estimate the independent effects of each variable, leading to unstable regression coefficients. This instability reduces the explanatory power of individual predictors and may result in misleading conclusions. Consequently, risk factors exhibiting a VIF > 10 were excluded from the analysis (Appendix A). Stratified analyses were conducted to account for dementia subtypes (AD and VD), baseline age groups (<65 years and ≥65 years), sex (male and female), and follow-up duration categories (≥5 years and ≥10 years). In subgroups where age and sex could serve as covariates, these variables were included as covariates to ensure robust results.

Variables that were statistically significant in the univariate Cox analysis for ACD and met the Schoenfeld residual test criteria were retained. To maintain consistency in the direction of the association between each risk factor and dementia, variables identified as favorable factors (HR < 1) in the univariate Cox analysis were reverse-coded to represent adverse factors. Risk factors within each domain were entered into a model and mutually adjusted, with age and sex included as covariates, and summed to create a domain-specific risk score, where higher scores indicated greater exposure to risk factors within that domain. These scores were further divided into tertiles, representing favorable, intermediate, and adverse exposure. However, due to an extremely uneven distribution, some data could not be divided into tertiles. In such cases, we aimed to minimize the discrepancy in participant numbers between the favorable, intermediate, and unfavorable groups as much as possible. The final divisions are detailed in Appendix A. To mitigate potential reverse causality, a follow-up period exceeding six years was established. A shorter follow-up period may fail to accurately capture the temporal relationship between risk factors and the onset of the disease, potentially leading to an overestimation of their impact. By extending the follow-up duration, we aimed to ensure a more precise assessment of the causal relationships between the identified risk factors and the development of the disease. The analysis approach for AD and VD followed the same methodology as for ACD. For psychiatric factors related to AD, only one variable reached significance. Therefore, it was categorized into favorable and adverse groups, resulting in a weight of zero, or a weighted value.

Univariate Cox models were used to analyze the relationships between seven domains and ACD, AD, and VD, with adjustments for age and sex (Model 1). Subsequently, multivariate Cox models were used to examine the associations between the seven risk factor domains and ACD, AD, and VD, also adjusting for age and sex (Model 2). To test the interactions between the seven domains and age or sex, terms of interaction with age and sex for each of the seven domains were included in the multivariable Cox analysis. Subsequently, subgroup analyses were performed separately, based on age and sex.

The PAF is defined as the proportion of patients with a specific disease that can be attributed to a particular risk factor. In other words, PAF measures the proportion of disease cases that could be prevented if a specific risk factor were eliminated. Consequently, this proportion of ACD, AD, and VD could be mitigated through the elimination of the risk factor. To ensure clarity in interpretation, binary variables were employed [36]. Initially, intermediate scores in the seven domains were converted to favorable scores, and the least favorable one-third was eliminated, thus yielding a more conservative estimate of dementia reduction (Model 1). Subsequently, unfavorable scores were transformed into intermediate scores, eliminating the least favorable two-thirds, providing a more comprehensive estimate of dementia reduction (Model 2). The relationship between dementia and exposure was modeled using logistic regression, with age and sex as covariates. The stdGlm function from the stdReg package facilitated the calculation of both factual and counterfactual prevalence rates, while the PAF was calculated using the formula in Equation (1) [37].
(1)$$\mathrm{PAF}=1\;-\;\frac{p\left(\mathrm{counterfactual}\;\mathrm{prevalence}\;\mathrm{rates}\right)}{p\left(\mathrm{factual}\;\mathrm{prevalence}\;\mathrm{rates}\right)}$$

The principal components were determined by calculating the eigenvalues and eigenvectors of the correlation matrix for the seven domains. The first two principal components were retained, and the communality for each domain was computed. The formula for calculating the weights of each domain is given in Equation (2).
(2)$$\mathrm{Weight}\;\mathrm{of}\;\mathrm{individual}\;\mathrm{domain}\;\left(w\right)=1\;-\;\mathrm{Communality}\;\mathrm{of}\;\mathrm{individual}\;\mathrm{domain}$$

The formula for calculating the overall PAF after incorporating the weights is given in Equation (3).
(3)$$o\mathrm{verall}\;\mathrm{PAF}=1-\left[\left(1\;-\;w_{1}\times {\mathrm{PAF}}_{1}\right)\left(1\;-\;w_{2}\;\times {\mathrm{PAF}}_{2}\right)\ldots \left(1-w_{7}\times {\mathrm{PAF}}_{7}\right)\right]$$

The formula for calculating the PAF of individual domain after incorporating the weights is given in Equation (4) [38].
(4)$$\mathrm{Weighted}\;\mathrm{PAF}\;\mathrm{of}\;\mathrm{individual}\;\mathrm{domain}=\frac{\;\mathrm{PAF}\;\mathrm{of}\;\mathrm{individual}\;\mathrm{domain}}{\Sigma \left(\mathrm{PAF}\;\mathrm{of}\;\mathrm{individual}\;\mathrm{domain}\right)}\times \mathrm{Overall}\;\mathrm{PAF}$$

All statistical tests were two-sided, and analyses were performed using R version 4.3.3 and Python version 3.9.

## 3. Results

The participant selection process is illustrated in Figure 1. Among the initial 502,369 UKB participants, 228 individuals with baseline dementia, 39,794 with more than 20% missing data, and 2778 with missing covariates were excluded, resulting in a final cohort comprising 460,799 participants. Within this cohort, 6849 participants were diagnosed with ACD, 2902 with AD, and 1525 with VD. The average age of all participants was 58 years, with a predominant representation of females. Notably, the likelihood of dementia development was observed to be higher among males and older individuals (refer to Appendix A Appendix A for details).

### 3.1. Risk Factors for Dementia

Among the 83 risk factors analyzed, 46 were found to be significantly associated with ACD. Of these, 21 factors demonstrated protective effects, while 25 factors exhibited harmful effects (Figure 2). Two of the top five factors were associated with a reduced risk of dementia. Current drinking was associated with an HR of 0.59, indicating a 41% reduction in the risk of developing dementia compared to that of non-drinkers. The 95% confidence interval (CI) ranged from 0.55–0.63, and the association was highly significant, with a *p*-value of 3.40 × 10^−51^. High handgrip strength had an HR of 0.64 (95% CI = 0.60–0.69, *p* = 7.15 × 10^−39^). Conversely, three factors increasing the risk of dementia were identified. Diabetes was associated with an HR of 2.63 (95% CI = 2.41–2.87, *p* = 7.34 × 10^−103^), making it one of the strongest risk factors. Hypertension showed an HR of 1.74 (95% CI = 1.64–1.85, *p* = 3.80 × 10^−73^), and stroke had an HR of 2.24 (95% CI = 1.92–2.60, *p* = 1.83 × 10^−25^).

Among the 83 risk factors analyzed, 18 were found to be significantly associated with AD. Of these, 10 factors demonstrated protective effects, while 8 factors exhibited harmful effects (Figure 3). Of the top five factors, three were associated with a reduced risk of AD: current drinking (HR = 0.62, 95% CI = 0.56–0.69, *p* = 7.38 × 10^−18^), unemployment (HR = 0.69, 95% CI = 0.62–0.75, *p* = 1.85 × 10^−15^), and high socioeconomic status (HR = 0.64, 95% CI = 0.55–0.73, *p* = 2.75 × 10^−10^). Two factors increased the risk of AD: diabetes (HR = 2.28, 95% CI = 1.98–2.63, *p* = 5.11 × 10^−30^) and hypertension (HR = 1.49, 95% CI = 1.35–1.64, *p* = 6.65 × 10^−16^).

Among the 83 risk factors analyzed, 41 were found to be significantly associated with VD. Of these, 19 factors demonstrated protective effects, while 22 factors exhibited harmful effects (Figure 3). All of the top five factors increased the risk of VD: diabetes (HR = 4.21, 95% CI = 3.62–4.89, *p* = 4.08 × 10^−79^), hypertension (HR = 2.74, 95% CI = 2.45–3.07, *p* = 1.12 × 10^−69^), stroke (HR = 5.06, 95% CI = 4.08–6.26, *p* = 7.28 × 10^−50^), high hemoglobin A1c (HbA1c) level (HR = 1.71, 95% CI = 1.54–1.89, *p* = 8.35 × 10^−25^), and coronary artery disease (HR = 2.58, 95% CI = 2.09–3.17, *p* = 3.62 × 10^−19^).

Additionally, when the results were stratified by dementia subtype, age, gender, and follow-up time, they remained consistent (Figure 3), suggesting that the identified associations are robust and not confounded by these variables. Interestingly, high socioeconomic status and unemployment did not meet the Schoenfeld residuals test for ACD, but did meet the requirments in specific subgroups, indicating potential modifying effects of age and dementia status. Moreover, alcohol use disorder did not meet the Schoenfeld residuals test for ACD, but showed a significant association in specific subgroups, suggesting that the relationship between alcohol and dementia risk may vary by dementia subtype, gender, and follow-up time.

### 3.2. Joint Effects of Risk Factors on Dementia

In the comprehensive analysis of dementia risk factors across seven domains, the study delineates a clear pattern of increased risk associated with intermediate and unfavorable factors compared to their favorable counterparts (Model 2, Table 1). Specifically, HR and their corresponding CI demonstrated significant associations. The top three unfavorable risk domains significantly increasing the risk of ACD are: medical history (HR = 1.88, 95% CI = 1.74–2.03, *p* < 2 × 10^−16^), psychiatric factors (HR = 1.61, 95% CI = 1.38–1.89, *p* = 4.08 × 10^−9^), and blood marker factors (HR = 1.44, 95% CI = 1.35–1.54, *p* < 2 × 10^−16^). The top three unfavorable risk domains significantly increasing the risk of AD are: medical history (HR = 1.80, 95% CI = 1.54–2.10, *p* = 9.91 × 10^−14^), sociodemographic factors (HR = 1.65, 95% CI = 1.46–1.87, *p* = 1.32 × 10^−15^), and psychiatric factors (HR = 1.50, 95% CI = 1.18–1.91, *p* = 0.00092). The top three unfavorable risk domains significantly increasing the risk of VD are: medical history (HR = 2.39, 95% CI = 2.10–2.71, *p* < 2 × 10^−16^), psychiatric factors (HR = 1.75, 95% CI = 1.21–2.53, *p* = 0.0027), and blood marker factors (HR = 1.61, 95% CI = 1.37–1.90, *p* = 1.17 × 10^−8^). Additional consistency was observed in the univariate analysis (Model 1, Appendix A).

Further stratified analyses based on age and gender revealed nuanced impacts. In the age subgroup analysis, certain unfavorable factors exerted a more pronounced impact on younger individuals. For example, unfavorable lifestyles (HR = 1.52, 95% CI = 1.38–1.67, *p* < 2 × 10^−16^) and medical history (HR = 2.09, 95% CI = 1.85–2.36, *p* < 2 × 10^−16^) significantly increase the risk of ACD among those under 65 (Table 2), while unfavorable sociodemographic factors (HR = 1.86, 95% CI = 1.58–2.20, *p* = 2.92 × 10^−13^) increased the risk of AD among the same age group (Table 3). Additionally, unfavorable blood marker factors (HR = 1.97, 95% CI = 1.50–2.60, *p* = 1.46 × 10^−6^) and unfavorable medical history (HR = 3.15, 95% CI = 2.56–3.89, *p* < 2 × 10^−16^) increased the risk of VD among those under 65 (Table 4).

In the gender subgroup analysis, certain unfavorable factors exerted a more pronounced impact on women. Notably, unfavorable sociodemographic factors (HR = 1.24, 95% CI = 1.14–1.34, *p* = 7.33 × 10^−7^) and unfavorable medical history (HR = 2.19, 95% CI = 1.95–2.46, *p* < 2 × 10^−16^) were particularly impactful for women in terms of ACD (Table 2). Similarly, unfavorable medical history (HR = 2.28, 95% CI = 1.84–2.82, *p* = 4.37 × 10^−14^) showed a significant association with AD among women (Table 3).

### 3.3. PAF Estimates for the Seven Domains in Dementia Prevention

The study explores two models (Model 1 and Model 2) aimed at preventing ACD, AD, and VD by transforming scores across seven domains. Model 1 focuses on converting intermediate scores to favorable ones, potentially achieving a notable reduction in ACD incidence by preventing 34.2% of cases (Table 5). This estimation highlights the substantial impact that modest enhancements in risk factors can exert on reducing dementia risk. Model 2 represents a more proactive strategy, aiming to convert unfavorable scores to intermediate levels, potentially preventing 56.2% of ACD cases. This underscores the critical role of early intervention and the substantial advantages of addressing risk factors proactively before they escalate. In the conservative scenario (Model 1), blood markers are identified as having the most substantial impact on ACD prevention, with a 9.3% reduction in incidence. This finding is particularly noteworthy, given the accessibility and modifiability of blood marker levels through lifestyle changes and medical interventions. The preventable range for AD spans from 48.5% to 59.2%, with psychiatric factors playing a key role (21.8% reduction) in the conservative scenario (Table 6). Similarly, the preventable range for VD is extensive, spanning 59.3–74.6%. Blood marker factors are again highlighted as having a significant impact, with a 15.3% reduction in VD incidence in the conservative scenario (Table 7).

## 4. Discussion

Among the 83 identified risk factors, significant association was observed with ACD and its subtypes: 46 factors were linked to ACD, 18 to AD, and 41 to VD. Certain risk factors exhibited similar trends across these dementia types. For example, individuals of medium socioeconomic status, those with high education levels, current drinkers, and those with high handgrip strength were found to have a reduced risk of developing dementia. Conversely, factors such as moderate levels of loneliness, elevated glucose levels, hypertension, diabetes, and thyroid dysfunction were associated with an increased risk. This pattern of risk and protective factors echoes findings from prior research, where similar attributes were recognized to influence the risk of dementia [39,40,41,42]. For example, research has demonstrated that higher education levels may help mitigate mild cognitive impairment and cognitive decline [43,44], both of which are significantly associated with dementia [45]. This protective effect is likely due to enhanced cognitive reserve [46,47], which can delay brain pathology and in turn, potentially reduce the risk of dementia [48,49]. Furthermore, regarding lifestyle factors, light to moderate alcohol consumption in middle-aged and older adults has been associated with a reduced risk of cognitive impairment and dementia [50], potentially due to cardiovascular protective mechanisms, such as favorable effects on lipid levels [51]. Meanwhile, the detrimental impact of chronic conditions like diabetes and hypertension on cognitive function has been well-documented [52,53], with these conditions often exacerbating the progression of dementia [53,54,55]. The presence of these factors underscores the multifaceted nature of dementia risk, with both lifestyle and health-related elements playing a role in its development. The majority of risk factors showed a consistent effect on the risk of dementia and its subtypes, with one significant exception: a high level of C-reactive protein. While a high C-reactive protein level is generally an indicator of increased risk due to its role as an inflammatory marker [56], in the context of AD, it was paradoxically associated with a decreased risk. This anomaly could be attributed to the neuroprotective role of neuroinflammation during the early or preclinical stages of AD, where microglia may mitigate the effects by degrading and clearing amyloid-beta (Aβ) [57] plaques, a hallmark of AD. In contrast, high-sensitivity-C-reactive protein is recognized as a risk biomarker for cerebrovascular disease [58], which can lead to pathological damage and cognitive impairment [59], thus increasing the risk of developing VD. This highlights the complex and multifaceted nature of dementia risk factors, underscoring the importance of considering both the specific type of dementia and the broader context of each risk factor.

In our analysis of dementia and its subtypes, we utilized HR and PAF as key metrics to evaluate risk domains. HR plays a pivotal role in quantifying the relative risk of an event between two groups over time, providing clarity regarding direct associations between risk domains and outcomes, and particularly in delineating risk disparities among different groups. Specifically, within our study, we found that unfavorable medical history exhibited the highest HR, followed by psychiatric factors and blood markers, significantly increasing the risk of ACD. Medical history also posed the greatest risk for AD, followed by sociodemographic and psychiatric factors. Similarly, for VD, medical history had the most significant impact, followed by psychiatric factors and blood markers. The prominence of medical history as a risk domain can be attributed to its encompassing a range of chronic conditions and health events that cumulatively contribute to the neuropathological changes associated with dementia. Chronic conditions like hypertension, diabetes, and heart disease can lead to vascular damage, which in turn impairs blood flow to the brain and contributes to neurodegeneration. Furthermore, long-standing medical issues often result in prolonged inflammation and oxidative stress, both of which are key mechanisms in the pathogenesis of dementia. 

While medical history stands as the most significant risk domain for dementia and its subtypes, the specific risk factors within this domain are not entirely uniform across different types of dementia. For example, risk factors associated with the medical history of AD, such as hypertension, diabetes, and thyroid dysfunction, are also encompassed within the risk domains of ACD and VD. Similarly, medical history risk factors related to VD, including stroke, atrial fibrillation, traumatic brain injury, anemia, and depression, are also present within the risk domain of ACD. This highlights the interconnected nature of these conditions and the multifactorial pathways through which chronic medical issues influence the development and progression of dementia. Rheumatoid arthritis is exclusively associated with ACD, potentially suggesting a connection with other dementia subtypes beyond AD and VD. Chronic inflammation resulting from rheumatoid arthritis triggers the release of cytokines that are linked to neuroinflammation, which can potentially lead to cognitive decline [60]. Moreover, the cardiovascular comorbidities often observed in rheumatoid arthritis patients, driven by shared inflammatory mechanisms, may exacerbate the risk of dementia [61]. Additionally, systemic inflammation associated with rheumatoid arthritis can increase the permeability of the blood–brain barrier, thereby facilitating neurodegeneration [62]. Furthermore, our findings suggest that women under the age of 65 are more susceptible to developing ACD, due to a predisposing unfavorable medical history. This heightened susceptibility can be attributed in part to hormonal changes, specifically the decline in estrogen levels during menopause, which have been linked to increased inflammation and vascular risks [63,64,65]. These factors significantly contribute to the pathogenesis of ACD.

Conversely, PAF estimates the proportion of cases that could potentially be prevented in the population if specific risk domains were eliminated. In this context, we focus on the risk domains that significantly contribute to PAF when the adverse aspects of the seven domains are completely eliminated (Model 2). PAF depends on both the distribution of exposure in the total population and the strength of the association between the risk domain and dementia, while HR reflects only the strength of the association between the risk domain and dementia. Due to the merging of exposure levels and uneven distribution, the risk domains most significantly contributing to PAF for dementia and its subtypes differ from those with the highest HR. For example, an unfavorable medical history is the most impactful risk domain for ACD, AD, and VD. However, its contribution to PAF is not the largest after data aggregation, possibly due to its relatively low exposure rate in dementia and its subtypes (exposure rate < 15%). Similarly, for AD, psychiatric factors make the greatest contribution to PAF, with sociodemographic factors and physical measures also playing significant roles. The close association between unfavorable psychiatric factors and AD, along with their high exposure rate (exposure rate > 90%), may explain this result. By identifying modifiable risk domains, PAF offers a broader perspective, aiding in the formulation of prevention strategies at the population level. When the unfavorable aspects of the seven domains are completely eliminated, blood markers emerge as the most significant contributors to the PAF of ACD. This is followed by psychiatric factors and lifestyle choices. Similarly, for VD, blood markers are the primary contributors to the PAF, followed by sociodemographic factors and medical history. The distribution of unfavorable blood markers in the population and their associations with dementia collectively identify them as the predominant risk domains for increasing the incidence of ACD and VD. These markers encompass various blood indicators reflecting overall health status, impacting dementia onset through oxidative stress, inflammation, and disruptions in biochemical pathways in peripheral tissues [66]. Most blood biomarkers are associated with both ACD and VD, such as medium levels of C-reactive protein and testosterone, high levels of vitamin D and glucose, medium and high levels of cystatin C, low-density lipoprotein (LDL), and HbA1c. These reflect the shared role of the biomarkers of different types of dementia in overall health status and disease mechanisms. Specific biomarkers, such as high eGFR, high alanine aminotransferase level, high urate level, and medium glucose level, are associated with ACD. Additionally, medium IGF-1 levels reduce the risk of VD compared to that associated with low IGF-1 levels. Higher concentrations of IGF-1 are correlated with larger brain volumes, suggesting a protective role against neurodegenerative diseases [67]. Moreover, IGF-1 plays a critical role in normal brain development. Deficiency in IGF-1 during childhood and adolescence can lead to growth retardation and cognitive impairments, which can be reversed by IGF-1 injection [68,69]. Furthermore, IGF-1 can cross the blood–brain barrier into the cerebrospinal fluid and brain parenchyma, thereby synchronously increasing in concentration with brain IGF-1 levels [70]. IGF-1 is produced by various brain cells [71], with specific brain regions involved in emotion, cognition, and memory, such as the hippocampus, amygdala, and thalamus, showing higher expression of IGF-1 receptor proteins [72]. This indicates that reducing IGF-1 levels may lead to dysfunction in these regions, further contributing to memory impairment and cognitive decline.

This study represents one of the most comprehensive investigations into the relationships between risk factors and dementia, including its subtypes, when compared to previous research. After first examining individual risk factors and their differential associations with ACD, AD, and VD, we then explored the variations in relationships between risk domains and these dementia types. However, several limitations must be acknowledged. Firstly, the generalizability of our findings is limited by the demographic composition of the UKB cohort, which predominantly consists of white British participants. This raises questions about the applicability of our results to other populations and ethnic groups. Differences in genetic backgrounds, environmental exposures, and lifestyle factors across various ethnicities and regions could significantly influence the risk factors for dementia. Secondly, the UKB sample may be biased towards a healthier population. The principle of voluntary participation suggests that individuals with severe health issues or unhealthy lifestyles may not participate in the study, potentially underestimating the impact of these risk factors on the broader population. Thirdly, the reliance on self-reported data introduces the potential for recall bias and inaccuracies, which could affect the validity of our findings. These self-reported measures may not accurately capture all relevant risk factors, thereby limiting the precision of our conclusions.

## 5. Conclusions

Dementia, a debilitating neurodegenerative condition, affects millions worldwide and poses significant challenges to public health. This study significantly advances our understanding of the associations between modifiable risk factors and dementia subtypes, leveraging a robust dataset from the UKB. By analyzing a wide range of risk factors, we identified critical domains that contribute to ACD, AD, and VD. Our findings underscore the potential of targeted interventions, particularly in managing medical history, to mitigate the risk of dementia. Additionally, our study highlights the differing impacts of risk domains, with blood markers playing a crucial role in ACD and VD prevention, and psychiatric factors being paramount for AD prevention. These insights provide a nuanced perspective that can guide public health strategies aimed at reducing dementia incidence. To effectively translate these findings into actionable public health initiatives and clinical protocols, we recommend the implementation of targeted screening programs designed to identify and quantify populations at higher risk. In parallel, the development of tailored intervention strategies, specifically designed to address the distinct risk factors identified in our study, is likely to enhance the effectiveness of prevention efforts. Moreover, it is essential to realign health education campaigns to emphasize the importance of early diagnosis and continuous monitoring, particularly for vulnerable groups. By doing so, we aim to reduce the overall burden of dementia and improve outcomes for those at greatest risk. This study not only advances our understanding of dementia prevention but also highlights the potential for targeted interventions to mitigate the global burden of dementia. The insights gained from this research are poised to inform the development of innovative strategies aimed at reducing the incidence and impact of dementia worldwide, marking a significant advancement in the field.

## Figures and Tables

**Figure 1 biomedicines-12-01967-f001:**
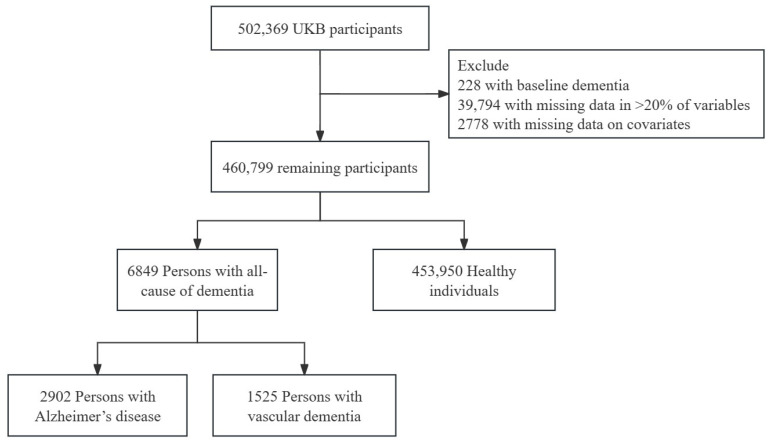
Flowchart illustrating the subject screening process.

**Figure 2 biomedicines-12-01967-f002:**
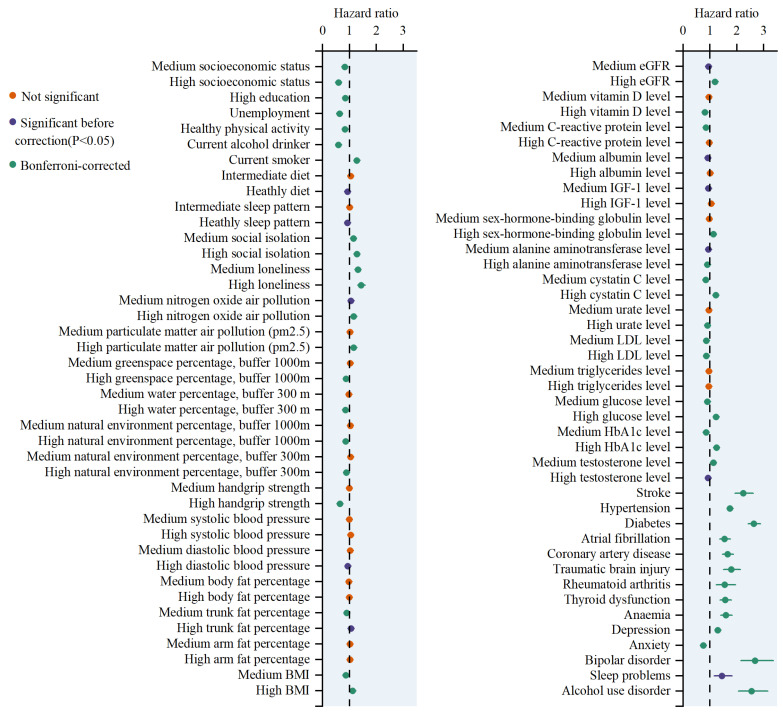
The association between various modifiable risk factors and the onset of ACD. Note: BMI, body mass index; LDL, low-density lipoprotein; HbA1c, hemoglobin A1c. The points represent HRs, with the horizontal lines indicating the corresponding 95% CIs. These HRs were calculated using Cox proportional hazards regression analysis, adjusted for baseline age and gender.

**Figure 3 biomedicines-12-01967-f003:**
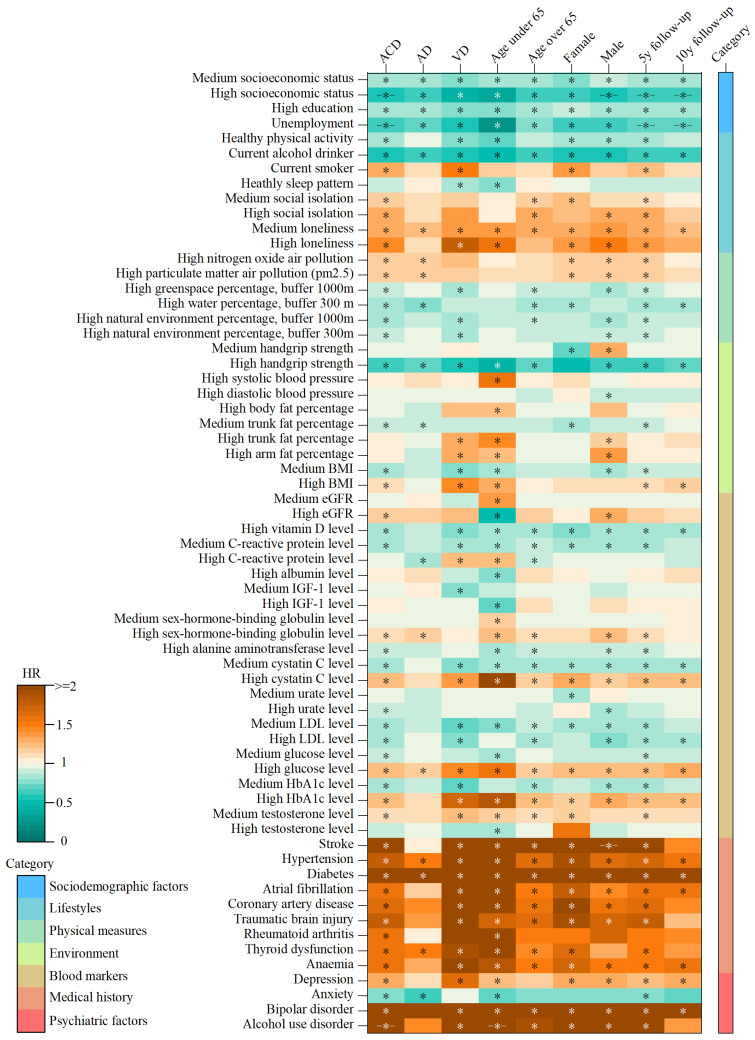
The heatmap visualizes the significant factors associated with ACD and its subgroups. Note: The model is adjusted for baseline age and gender. The cell colors represent the effect sizes (HR) of the risk factors pertinent to each study group. Asterisks within the cells indicate significant associations following Bonferroni correction. Short horizontal lines preceding and following the asterisks denote factors that did not pass the Schoenfeld residual test.

**Table 1 biomedicines-12-01967-t001:** Associations of the seven domains with dementia and its subtypes (Model 2).

Domains	ACD			AD			VD		
	HR (95% CI)	*p*	*p* for Trend	HR (95% CI)	*p*	*p* for Trend	HR (95% CI)	*p*	*p* for Trend
Sociodemographic factors									
Favorable	1(reference)		***	1(reference)		***	1(reference)		***
Intermediate	1.08(0.99–1.17)	.		1.27(1.12–1.45)	***		1.21(1.02–1.45)	*	
Unfavorable	1.15(1.09–1.22)	***		1.65(1.46–1.87)	***		1.58(1.34–1.86)	***	
Lifestyles									
Favorable	1(reference)		***	1(reference)		***	1(reference)		***
Intermediate	1.16(1.08–1.23)	***		1.17(1.07–1.29)	**		1.13(0.98–1.29)	.	
Unfavorable	1.37(1.29–1.46)	***		1.43(1.27–1.62)	***		1.49(1.30–1.70)	***	
Environment									
Favorable	1(reference)		***	1(reference)		***	1(reference)		*
Intermediate	1.11(1.04–1.18)	**		1.19(1.08–1.32)	***		1.20(1.01–1.42)	*	
Unfavorable	1.15(1.08–1.22)	***		1.25(1.13–1.38)	***		1.19(1.03–1.37)	*	
Physical measures									
Favorable	1(reference)		***	1(reference)		***	1(reference)		**
Intermediate	1.25(1.17–1.33)	***		1.14(1.00–1.30)	*		1.19(1.01–1.40)	*	
Unfavorable	1.18(1.09–1.28)	***		1.28(1.13–1.44)	***		1.23(1.07–1.40)	**	
Blood markers									
Favorable	1(reference)		***	1(reference)		***	1(reference)		***
Intermediate	1.19(1.10–1.27)	***		1.33(1.20–1.48)	***		1.15(0.96–1.38)		
Unfavorable	1.44(1.35–1.54)	***		1.29(1.17–1.42)	***		1.61(1.37–1.90)	***	
Medical history									
Favorable	1(reference)		***	1(reference)		***	1(reference)		***
Intermediate	1.29(1.20–1.40)	***		1.27(1.13–1.43)	***		1.56(1.25–1.95)	***	
Unfavorable	1.88(1.74–2.03)	***		1.80(1.54–2.10)	***		2.39(2.10–2.71)	***	
Psychiatric factors									
Favorable	1(reference)		***	1(reference)		***	1(reference)		***
Intermediate	1.23(1.07–1.42)	**					1.47(1.25–1.72)	***	
Unfavorable	1.61(1.38–1.89)	***		1.50(1.18–1.91)	***		1.75(1.21–2.53)	**	

Note: Statistical significance levels are indicated as follows: *** for *p*-values between 0 and 0.001; ** for *p*-values between 0.001 and 0.01; * for *p*-values between 0.01 and 0.05; “.” for *p*-values between 0.05 and 0.1. In each domain, the favorable condition was used as the reference. Associations were estimated using a Cox proportional hazards model incorporating all seven domains, adjusted for age and gender.

**Table 2 biomedicines-12-01967-t002:** Associations of the seven domains with ACD in age and sex subgroups.

Domains	Age < 65		Age ≥ 65		*p* ^1^	Female (Aged 56.3 ± 8.0)		Male (Aged 56.7 ± 8.2)		*p* ^2^
	HR (95% CI)	*p*	HR (95% CI)	*p*		HR (95% CI)	*p*	HR (95% CI)	*p*	
Sociodemographic factors										
Favorable	1 (reference)		1 (reference)			1 (reference)		1 (reference)		
Intermediate	1.06 (0.93–1.20)		1.10 (0.99–1.22)	.		1.05 (0.93–1.18)		1.12 (1.00–1.26)	*	
Unfavorable	1.20 (1.10–1.31)	***	1.13 (1.05–1.22)	**		1.24 (1.14–1.34)	***	1.08 (1.00–1.17)	*	*
Lifestyles										
Favorable	1 (reference)		1 (reference)			1 (reference)		1 (reference)		
Intermediate	1.23 (1.11–1.37)	***	1.10 (1.01–1.20)	*	*	1.16 (1.05–1.27)	**	1.15 (1.05–1.26)	**	
Unfavorable	1.52 (1.38–1.67)	***	1.26 (1.17–1.37)	***	***	1.39 (1.27–1.52)	***	1.34 (1.23–1.46)	***	
Environment										
Favorable	1 (reference)		1 (reference)			1 (reference)		1 (reference)		
Intermediate	1.03 (0.93–1.15)		1.15 (1.05–1.25)	**		1.10 (1.00–1.21)	*	1.10 (1.01–1.21)	*	
Unfavorable	1.14 (1.04–1.25)	**	1.15 (1.06–1.24)	***		1.22 (1.12–1.33)	***	1.08 (1.00–1.18)	.	
Physical measures										
Favorable	1 (reference)		1 (reference)			1 (reference)		1 (reference)		
Intermediate	1.34 (1.21–1.49)	***	1.20 (1.10–1.30)	***		1.15 (1.01–1.30)	*	1.32 (1.22–1.43)	***	
Unfavorable	1.28 (1.13–1.45)	***	1.13 (1.02–1.25)	*		1.09 (0.95–1.25)		1.19 (1.06–1.33)	**	
Blood markers										
Favorable	1 (reference)		1 (reference)			1 (reference)		1 (reference)		
Intermediate	1.25 (1.12–1.39)	***	1.13 (1.03–1.24)	*		1.17 (1.05–1.30)	**	1.21 (1.10–1.33)	***	
Unfavorable	1.60 (1.44–1.78)	***	1.33 (1.22–1.46)	***	.	1.40 (1.27–1.55)	***	1.50 (1.37–1.64)	***	
Medical history										
Favorable	1 (reference)		1 (reference)			1 (reference)		1 (reference)		
Intermediate	1.48 (1.31–1.67)	***	1.20 (1.09–1.32)	***	**	1.36 (1.22–1.52)	***	1.23 (1.11–1.37)	***	
Unfavorable	2.09 (1.85–2.36)	***	1.78 (1.61–1.95)	***	***	2.19 (1.95–2.46)	***	1.69 (1.53–1.87)	***	***
Psychiatric factors										
Favorable	1 (reference)		1 (reference)			1 (reference)		1 (reference)		
Intermediate	1.28 (1.04–1.57)	*	1.18 (0.96–1.44)			1.26 (1.05–1.53)	*	1.17 (0.94–1.47)		
Unfavorable	1.72 (1.38–2.16)	***	1.48 (1.18–1.86)	***	.	1.72 (1.39–2.11)	***	1.48 (1.15–1.89)	**	

Note: Statistical significance levels are indicated as follows: *** for *p*-values between 0 and 0.001; ** for *p*-values between 0.001 and 0.01; * for *p*-values between 0.01 and 0.05; “.” for *p*-values between 0.05 and 0.1. In each domain, the favorable condition was used as the reference. Associations were estimated using a Cox proportional hazards model incorporating all seven domains, adjusted for age and gender. *p*^1^, interaction between domain and age; *p*^2^, interaction between domain and sex.

**Table 3 biomedicines-12-01967-t003:** Associations of the seven domains with AD in age and sex subgroups.

Domains	Age < 65		Age ≥ 65		*p* ^1^	Female (Aged 56.3 ± 8.0)		Male (Aged 56.7 ± 8.2)		*p* ^2^
	HR (95% CI)	*p*	HR (95% CI)	*p*		HR (95% CI)	*p*	HR (95% CI)	*p*	
Sociodemographic factors										
Favorable	1 (reference)		1 (reference)			1 (reference)		1 (reference)		
Intermediate	1.50 (1.26–1.78)	***	0.99 (0.82–1.19)		***	1.21 (1.00–1.46)	*	1.34 (1.13–1.60)	**	
Unfavorable	1.86 (1.58–2.20)	***	1.31 (1.10–1.57)	**	***	1.64 (1.38–1.96)	***	1.67 (1.41–1.99)	***	
Lifestyles										
Favorable	1 (reference)		1 (reference)			1 (reference)		1 (reference)		
Intermediate	1.14 (0.98–1.32)		1.19 (1.05–1.34)	**		1.15 (1.00–1.31)	*	1.20 (1.05–1.37)	**	
Unfavorable	1.50 (1.22–1.83)	***	1.41 (1.21–1.64)	***		1.37 (1.17–1.61)	***	1.50 (1.23–1.83)	***	
Environment										
Favorable	1 (reference)		1 (reference)			1 (reference)		1 (reference)		
Intermediate	1.12 (0.96–1.31)		1.25 (1.10–1.41)	***		1.18 (1.03–1.35)	*	1.21 (1.06–1.39)	**	
Unfavorable	1.24 (1.06–1.46)	**	1.26 (1.11–1.44)	***		1.35 (1.18–1.55)	***	1.14 (0.99–1.33)	.	
Physical measures										
Favorable	1 (reference)		1 (reference)			1 (reference)		1 (reference)		
Intermediate	1.20 (0.97–1.48)	.	1.12 (0.95–1.32)			2.66 (0.66–10.68)		1.20 (1.03–1.40)	*	
Unfavorable	1.41 (1.16–1.71)	***	1.22 (1.05–1.43)	*		3.07 (0.77–12.31)		1.26 (1.10–1.44)	***	
Blood markers										
Favorable	1 (reference)		1 (reference)			1 (reference)		1 (reference)		
Intermediate	1.34 (1.14–1.58)	***	1.34 (1.17–1.52)	***		1.35 (1.18–1.54)	***	1.37 (1.15–1.64)	***	
Unfavorable	1.27 (1.09–1.48)	**	1.30 (1.15–1.47)	***		1.37 (1.19–1.57)	***	1.24 (1.09–1.42)	**	
Medical history										
Favorable	1 (reference)		1 (reference)			1 (reference)		1 (reference)		
Intermediate	1.38 (1.14–1.69)	**	1.22 (1.05–1.40)	**		1.32 (1.12–1.56)	***	1.21 (1.03–1.43)	*	
Unfavorable	1.88 (1.43–2.46)	***	1.79 (1.48–2.16)	***		2.28 (1.84–2.82)	***	1.44 (1.15–1.81)	**	**
Psychiatric factors										
Favorable	1 (reference)		1 (reference)			1 (reference)		1 (reference)		
Unfavorable	1.82 (1.26–2.64)	**	1.30 (0.95–1.78)			1.48 (1.10–2.00)	**	1.54 (1.03–2.30)	*	

Note: Statistical significance levels are indicated as follows: *** for *p*-values between 0 and 0.001; ** for *p*-values between 0.001 and 0.01; * for *p*-values between 0.01 and 0.05; “.” for *p*-values between 0.05 and 0.1. In each domain, the favorable condition was used as the reference. Associations were estimated using a Cox proportional hazards model incorporating all seven domains, adjusted for age and gender. *p*^1^, interaction between domain and age; *p*^2^, interaction between domain and sex.

**Table 4 biomedicines-12-01967-t004:** Associations of the seven domains with VD age and sex subgroups.

Domains	Age < 65		Age ≥ 65		*p* ^1^	Female (Aged 56.3 ± 8.0)		Male (Aged 56.7 ± 8.2)		*p* ^2^
	HR (95% CI)	*p*	HR (95% CI)	*p*		HR (95% CI)	*p*	HR (95% CI)	*p*	
Sociodemographic factors										
Favorable	1 (reference)		1 (reference)			1 (reference)		1 (reference)		
Intermediate	1.31 (1.00–1.71)	*	1.00 (0.80–1.27)			1.29 (0.96–1.73)	.	1.19 (0.95–1.48)		
Unfavorable	1.95 (1.54–2.47)	***	1.22 (0.98–1.51)	.	***	1.57 (1.20–2.06)	***	1.60 (1.30–1.97)	***	
Lifestyles										
Favorable	1 (reference)		1 (reference)			1 (reference)		1 (reference)		
Intermediate	1.08 (0.85–1.38)		1.15 (0.97–1.36)			1.23 (0.99–1.53)	.	1.05 (0.87–1.26)		
Unfavorable	1.59 (1.27–2.00)	***	1.39 (1.17–1.65)	***		1.54 (1.24–1.90)	***	1.44 (1.21–1.72)	***	
Environment										
Favorable	1 (reference)		1 (reference)			1 (reference)		1 (reference)		
Intermediate	1.14 (0.85–1.53)		1.24 (1.00–1.53)	*		1.07 (0.82–1.39)		1.32 (1.06–1.66)	*	
Unfavorable	1.19 (0.93–1.52)		1.19 (0.99–1.43)	.		1.11 (0.89–1.38)		1.26 (1.03–1.53)	*	
Physical measures										
Favorable	1 (reference)		1 (reference)			1 (reference)		1 (reference)		
Intermediate	1.04 (0.77–1.40)		1.27 (1.04–1.54)	*	*	0.95 (0.71–1.26)		1.31 (1.07–1.60)	**	
Unfavorable	1.34 (1.07–1.68)	*	1.18 (0.99–1.40)	.		0.98 (0.79–1.21)		1.42 (1.19–1.69)	***	**
Blood markers										
Favorable	1 (reference)		1 (reference)			1 (reference)		1 (reference)		
Intermediate	1.31 (0.97–1.77)	.	1.05 (0.84–1.31)		.	0.94 (0.72–1.23)		1.37 (1.07–1.75)	*	.
Unfavorable	1.97 (1.50–2.60)	***	1.41 (1.15–1.72)	***	*	1.51 (1.19–1.92)	***	1.74 (1.38–2.18)	***	
Medical history										
Favorable	1 (reference)		1 (reference)			1 (reference)		1 (reference)		
Intermediate	1.71 (1.18–2.49)	**	1.48 (1.12–1.95)	**		1.48 (1.04–2.11)	*	1.64 (1.23–2.18)	***	
Unfavorable	3.15 (2.56–3.89)	***	2.06 (1.76–2.41)	***	***	2.61 (2.14–3.17)	***	2.26 (1.91–2.66)	***	
Psychiatric factors										
Favorable	1 (reference)		1 (reference)			1 (reference)		1 (reference)		
Intermediate	1.41 (1.11–1.81)	**	1.48 (1.20–1.82)	***		1.65 (1.33–2.06)	***	1.32 (1.05–1.67)	*	
Unfavorable	1.97 (1.23–3.16)	**	1.30 (0.71–2.36)	***		4.32 (2.57–7.26)	***	1.05 (0.63–1.76)	*	***

Note: Statistical significance levels are indicated as follows: *** for *p*-values between 0 and 0.001; ** for *p*-values between 0.001 and0.01; * for *p*-values between 0.01 and 0.05; “.” for *p*-values between 0.05 and 0.1. In each domain, the favorable condition was used as the reference. Associations were estimated using a Cox proportional hazards model incorporating all seven domains, adjusted for age and gender. *p*^1^, interaction between domain and age; *p*^2^, interaction between domain and sex.

**Table 5 biomedicines-12-01967-t005:** Weighted and unweighted PAF of ACD for the seven domains.

Domains	Model 1			Model 2		
	Unweighted PAF	Communality	Weighted_ PAF	Unweighted PAF	Communality	Weighted_ PAF
Sociodemographic factors	0.097	0.016	0.059	0.118	0.185	0.059
Lifestyle	0.123	0.277	0.075	0.198	0.183	0.099
Environment	0.050	0.034	0.031	0.086	0.095	0.043
Physical measures	0.035	0.190	0.021	0.148	0.388	0.074
Blood markers	0.152	0.423	0.093	0.242	0.349	0.121
Medical history	0.080	0.404	0.049	0.119	0.292	0.059
Psychiatric factors	0.022	0.658	0.014	0.216	0.509	0.108
Overall PAF			0.342			0.562

**Table 6 biomedicines-12-01967-t006:** Weighted and unweighted PAF of AD for the seven domains.

Domains	Model 1			Model 2		
	Unweighted PAF	Communality	Weighted_ PAF	Unweighted PAF	Communality	Weighted_ PAF
Sociodemographic factors	0.177	0.287	0.112	0.292	0.394	0.155
Lifestyle	0.045	0.291	0.029	0.099	0.159	0.052
Environment	0.054	0.056	0.034	0.090	0.008	0.048
Physical measures	0.082	0.321	0.052	0.137	0.460	0.072
Blood markers	0.029	0.497	0.018	0.091	0.465	0.048
Medical history	0.036	0.407	0.023	0.065	0.383	0.035
Psychiatric factors	0.344	0.141	0.218	0.344	0.131	0.183
Overall PAF			0.485			0.592

**Table 7 biomedicines-12-01967-t007:** Weighted and unweighted PAF of VD for the seven domains.

Domains	Model 1			Model 2		
	Unweighted PAF	Communality	Weighted_ PAF	Unweighted PAF	Communality	Weighted_ PAF
Sociodemographic factors	0.224	0.186	0.117	0.322	0.268	0.144
Lifestyle	0.161	0.305	0.084	0.248	0.305	0.111
Environment	0.086	0.234	0.045	0.213	0.186	0.096
Physical measures	0.141	0.277	0.073	0.184	0.170	0.082
Blood markers	0.295	0.308	0.153	0.389	0.267	0.174
Medical history	0.220	0.300	0.114	0.255	0.311	0.115
Psychiatric factors	0.014	0.390	0.007	0.052	0.493	0.023
Overall PAF			0.593			0.746

## Data Availability

The datasets used in this study, sourced from the UK Biobank, can be accessed through the UK Biobank data access process, as outlined at http://www.ukbiobank.ac.uk/register-apply/ (accessed on 24 February 2023). The UK Biobank’s Research Access Administration Team manages all data access requests from both academic and commercial researchers impartially. Requests are thoroughly evaluated to ensure they align with public health research objectives and are promptly approved if they do.

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
