# Peer review of "Investigating Modifiable Risk Factors Across Dementia Subtypes: Insights from the UK Biobank"

_biomedicines, 2024, doi:10.3390/biomedicines12091967_

Round 1

Reviewer 1 Report

Comments and Suggestions for Authors

The ms. entitled “Investigating Modifiable Risk Factors Across Dementia Sub- 2 types: Insights from the UK Biobank” aims to investigate the relationship between modifiable risk factors and dementia subtypes. An important aspect of the study was calculating the population-attributable fractions (PAF) for each risk domain. PAF was utilized to estimate the potential 16 impact of eliminating adverse characteristics of the risk domains. The result revealed that an unfavorable medical history significantly increased the risk of all causes of dementia (ACD), Alzheimer's disease (AD), and vascular dementia (VD).

The paper would provide a significant contribution to this research field. This study underscores the potential for preventing dementia and its subtypes through targeted interventions on modifiable risk factors. The distinct insights provided by HR and PAF emphasize the importance of considering both the strength of associations and the population-level impact of dementia prevention strategies. Anyway, in my opinion, minor changes are needed to improve the quality of the ms. These issues need to be addressed by the authors.

METHODS

1)    The statistical methods section is highly technical and may benefit from a clearer explanation of some terms and processes for broader accessibility. Moreover, While the exclusion criteria are detailed, discussing potential biases introduced by these exclusions would strengthen the section. I encourage the authors to address this point.

2)    Please, I suggest inserting the mean and standard deviation of age split by gender.

RESULTS

3)    The result section is really well written.

DISCUSSION

4)    The results underscore the importance of lifestyle factors in mitigating the risk of dementia. I encourage the authors to highlight more clearly these aspects and the protective role of education and intellectual activities against MCI and cognitive decline in general. I suggest to consult the following articles: 10.3390/ijerph19053097; 10.1016/S2468-2667(20)30185-7

CONCLUSION

5)    The conclusion section is clear and well explained. However, the authors could provide more specific recommendations for public health policy and clinical practice based on the findings.

6)    The limitations are poorly explained. I recommend a more discussion of these in the section.

Author Response

Dear Editor and Reviewers,

   I would like to extend my sincere gratitude for your valuable time and constructive feedback on my manuscript. Your insights have significantly enhanced the quality of the work. I truly appreciate your dedication to the peer-review process.

Thank you for your contributions.

Best regards,

Lan Lin

Comments 1: [The statistical methods section is highly technical and may benefit from a clearer explanation of some terms and processes for broader accessibility. Moreover, While the exclusion criteria are detailed, discussing potential biases introduced by these exclusions would strengthen the section. I encourage the authors to address this point.]

Response 1: Thank you for underscoring this critical aspect. I fully agree with your observation and have accordingly expanded the explanation on the primary metric derived from the Cox model, the Population Attributable Fraction (PAF) and its definition. Additionally, I have provided a thorough analysis of the potential biases that could arise from not applying the Bonferroni correction, not addressing multicollinearity, and not setting the follow-up period to six years.

Comments 2: [ Please, I suggest inserting the mean and standard deviation of age split by gender.]

Response 2: Thank you for pointing this out. I agree with your suggestion and have accordingly included the mean age and standard deviation by gender in the first row of Tables 3, 4, and 5, as well as in the fifth and sixth rows of the first column in Appendix Table 6.

Comments 3: [The result section is really well written.]

Response 3: Thank you, I am very honored to receive your recognition.

Comments 4: [The results underscore the importance of lifestyle factors in mitigating the risk of dementia. I encourage the authors to highlight more clearly these aspects and the protective role of education and intellectual activities against MCI and cognitive decline in general. I suggest to consult the following articles: 10.3390/ijerph19053097; 10.1016/S2468-2667(20)30185-7.]

Response 4: Thank you for highlighting this point. I concur with your observation and have therefore provided a comprehensive description of the protective role of education in mitigating the risk of mild cognitive impairment, cognitive decline, and dementia. This includes an exploration of the underlying mechanisms, supported by relevant references (see lines 401-406 in red). Furthermore, regarding lifestyle factors, I have elaborated on how current alcohol consumption has demonstrated protective effects against both dementia and its subtypes, with a detailed explanation of the mechanisms involved (see lines 406-409 in red).

Comments 5: [The conclusion section is clear and well explained. However, the authors could provide more specific recommendations for public health policy and clinical practice based on the findings.]

Response 5: Thank you for pointing this out. I agree with this comment. Therefore, I have provided detailed strategies for dementia prevention in the conclusion section (lines 535-543 in red font).

Comments 6: [The limitations are poorly explained. I recommend a more discussion of these in the section.]

Response 6: Thank you for pointing this out. I agree with this comment. Therefore, I have added a description of the limitations in the last paragraph of the discussion section (lines 511-523 in red font).

Reviewer 2 Report

Comments and Suggestions for Authors

The introduction is very long and hard to be followed. It needs to be shortened according to the main aims of the study. The details of UKB is better to be mentioned in methods. 

A major concern is the big sample size, which may cause increasing statistical power and attaining more significant results. How authors deal with this? Several statistically significant results have been made, and this can be due to large sample size. 

Most of these risk factors have been previously studied and founded. Insisting on the novelty of the present study, considering the available literature, does not make sense and should be modified. 

Comments on the Quality of English Language

The manuscript needs to be revised in terms of English and medical writing. 

Author Response

Dear Editor and Reviewers,

   I would like to extend my sincere gratitude for your valuable time and constructive feedback on my manuscript. Your insights have significantly enhanced the quality of the work. I truly appreciate your dedication to the peer-review process.

Comments 1: [The introduction is very long and hard to be followed. It needs to be shortened according to the main aims of the study. The details of UKB is better to be mentioned in methods.]

Response 1: Thank you for highlighting this point. I fully agree with your observation. Consequently, I have revised the introduction by removing redundant details. Additionally, I have streamlined the final paragraph of the introduction to more effectively emphasize the research focus.

Comments 2: [A major concern is the big sample size, which may cause increasing statistical power and attaining more significant results. How authors deal with this? Several statistically significant results have been made, and this can be due to large sample size.]

Response 2: Thank you for raising this important issue. While it is true that a large sample size can enhance statistical power and potentially lead to more significant results, our approach to analyzing the data ensured that statistical significance was not our sole criterion. We also focused on effect sizes, specifically hazard ratios (HRs). Our analysis revealed that most unfavorable risk domains had HRs greater than 1.2, indicating a clinically significant association. Furthermore, several risk domains with HRs exceeding 1.5 demonstrated a substantial clinical impact. To rigorously address the concern of false positives, we employed a Bonferroni correction in our univariate Cox analysis. This adjustment revised the significance threshold to 0.0006024 (0.05 divided by 83 tests), thereby controlling the false positive rate and ensuring that only valid risk factors were selected for further analysis. Additionally, we conducted stratified univariate analyses using Cox models, with two separate models for each risk domain cox analysis and Population Attributable Fraction (PAF) analysis. Sensitivity analyses further corroborated the robustness and reliability of our findings. These steps collectively help to mitigate the impact of the large sample size and confirm the clinical relevance of the observed associations.

Comments 3: [Most of these risk factors have been previously studied and founded. Insisting on the novelty of the present study, considering the available literature, does not make sense and should be modified.]

Response 3: Thank you for highlighting this point. It is well recognized that dementia risk factors have been extensively studied over the past two decades, with significant contributions from the research community. Although many of these risk factors have been identified in previous studies, most research has focused on the impact of individual risk factors or small groups of factors. In contrast, our study leverages the extensive data available from the UK Biobank to address several gaps in the existing literature. Specifically, we not only evaluated the effects of individual risk factors on dementia and its subtypes but also examined the combined impact of multiple risk domains. Additionally, we quantified the potential reduction in dementia prevalence that could be achieved by mitigating these risk domains. By integrating a comprehensive analysis of both individual and combined risk factors, our study provides new insights and practical recommendations for dementia prevention. This approach offers a more holistic understanding of dementia risk, filling gaps left by previous research and contributing valuable knowledge to the field.

Reviewer 3 Report

Comments and Suggestions for Authors

I am grateful for the opportunity to review the work ‘’Investigating Modifiable Risk Factors Across Dementia Sub-2 types: Insights from the UK Biobank”. This is an interesting study that

I welcome the opportunity to review the paper ‘’. This is an interesting study that looks at the relationship between risk factors and different types of dementia. It uses a large sample of 460,799 participants. It uses univariate Cox proportional hazards regression models and examines associations between 83 risk factors and dementia risks in different conditions. Results indicate that unfavourable medical history significantly increases the risk of ACD, AD and VD. Also, blood markers were found to be the most important risk domain for preventing AD and psychiatric factors were found to be the most important risk domain for preventing AD.

However, some suggestions for improvement of the work will be presented below.

1. At the end of the introduction, it is recommended to clearly disaggregate the research objectives or hypotheses.

2. Likewise, in the section on it is recommended to divide it for a better understanding of the reader into the following sub-sections: participants (description of the sample, its selection), procedure (describe the phases and include a graph or a table in which they are clearly visible to improve the reader's understanding), instruments (clearly describe the instruments used, so that the work can be replicable), data analysis (clearly describe the statistics or tests used to contrast each of the hypotheses formulated, which is why it is important to first make the hypotheses explicit). You have all of this information, although arranging it in the way I suggest will make it easier for researchers to read.

3. Regarding the Results section, this should be presented in order with respect to the research hypotheses. Therefore, it is important to describe them clearly.

4. Regarding the sections you indicate that it is not applicable, when working with clinical data it is applicable. If you have collected these data from a databank, the prior informed consent of the patients and the approval of the bioethics committees of the institutions will appear there. Please reflect this.

Minor points, check the legends of tables and figures, they cannot be too long (e.g. Figure 2, Figure 3), if you need to include more information, please put a note.

Author Response

Dear Editor and Reviewers,

   I would like to extend my sincere gratitude for your valuable time and constructive feedback on my manuscript. Your insights have significantly enhanced the quality of the work. I truly appreciate your dedication to the peer-review process.

Comments 1: [At the end of the introduction, it is recommended to clearly disaggregate the research objectives or hypotheses.]

Response 1: Thank you for highlighting this point. I agree with this comment. Therefore, I have clearly divided the research objectives into three parts in the last paragraph of the introduction. 

Comments 2: [Likewise, in the section on it is recommended to divide it for a better understanding of the reader into the following sub-sections: participants (description of the sample, its selection), procedure (describe the phases and include a graph or a table in which they are clearly visible to improve the reader's understanding), instruments (clearly describe the instruments used, so that the work can be replicable), data analysis (clearly describe the statistics or tests used to contrast each of the hypotheses formulated, which is why it is important to first make the hypotheses explicit). You have all of this information, although arranging it in the way I suggest will make it easier for researchers to read.]

Response 2: Thank you for bringing this to my attention. I agree with your suggestion and have accordingly revised the study population section by dividing it into three categories: study population, dementia diagnoses, and modifiable risk factors, aligning with the content of this article. Additionally, I have included the tools used at the end of the statistical analysis section. To enhance the presentation of our findings, I have moved the process and results of screening the study population to the results section. I believe these changes will help clarify the methods section.

Comments 3: [Regarding the Results section, this should be presented in order with respect to the research hypotheses. Therefore, it is important to describe them clearly.]

Response 3: Thank you for pointing this out. I agree with this comment. Therefore, I have explicitly broken down the research objectives into three parts. So the order in which the results are presented is exactly the same as the order of the research objectives

Comments 4: [Regarding the sections you indicate that it is not applicable, when working with clinical data it is applicable. If you have collected these data from a databank, the prior informed consent of the patients and the approval of the bioethics committees of the institutions will appear there. Please reflect this.]

Response 4: We updated the manuscript to include a statement acknowledging that all necessary ethical approvals and informed consent procedures were adhered to, reflecting this in the revised sections of the text.

Comments 5: [Minor points, check the legends of tables and figures, they cannot be too long (e.g. Figure 2, Figure 3), if you need to include more information, please put a note.]

Response 5: Thank you for pointing this out. I agree with this comment. Therefore, I have put the extra information inside the title into the notes.

Round 2

Reviewer 2 Report

Comments and Suggestions for Authors

My concerns have been addressed well by the authors, and the paper looks good now. Thank you.